# Distinct skyrmion phases at room temperature in two-dimensional ferromagnet Fe₃GaTe₂

Xiaowei Lv[1,6], Hualiang Lv [2,6], Yalei Huang[3,6], Ruixuan Zhang[4], Guanhua Qin[4], Yihui Dong[3], Min Liu[1], Ke Pei[1], Guixin Cao [3,4] ✉, Jincang Zhang[4], Yuxiang Lai[5] & Renchao Che [1,4] ✉

Distinct skyrmion phases at room temperature hosted by one material offer additional degree of freedom for the design of topology-based compact and energetically-efficient spintronic devices. The field has been extended to low-dimensional magnets with the discovery of magnetism in two-dimensional van der Waals magnets. However, creating multiple skyrmion phases in 2D magnets, especially above room temperature, remains a major challenge. Here, we report the experimental observation of mixed-type skyrmions, exhibiting both Bloch and hybrid characteristics, in a room-temperature ferromagnet Fe₃GaTe₂. Analysis of the magnetic intensities under varied imaging conditions coupled with complementary simulations reveal that spontaneous Bloch skyrmions exist as the magnetic ground state with the coexistence of hybrid stripes domain, on account of the interplay between the dipolar interaction and the Dzyaloshinskii-Moriya interaction. Moreover, hybrid skyrmions are created and their coexisting phases with Bloch skyrmions exhibit considerably high thermostability, enduring up to 328 K. The findings open perspectives for 2D spintronic devices incorporating distinct skyrmion phases at room temperature.

Magnetic skyrmions[1,2] with topological nanoscale spin configurations have garnered increasing attention as a source of nontrivial emergent phenomena and promising candidates for application in spintronic storage such as racetrack memories[3,4]. In such a device, skyrmions are expected to move along the racetrack, encoding information through their presence or absence. However, the transverse motion of skyrmions in response to external electric currents leads to their annihilation at the track boundary, known as the skyrmion Hall effect[5–7], is regarded as a major obstacle in topology-based memories. Although there are several methods proposed to address this issue such as magnetization compensation using synthetic antiferromagnets[8–10] and ferrimagnets[11,12], one ferromagnetic layer as a preferable system for applications has not been considered. Recently, theoretical studies have suggested that a zero Hall angle can also be achieved by a hybrid skyrmion within ferromagnets, namely the mixed Bloch-Neel chiral spin texture[13,14]. This is further confirmed by the experimental observation of the intermediate skyrmions between the Bloch- and Neel-type in Ta/CoFeB/Ir heterostructures, where such skyrmions exhibit straight motion without the skyrmion Hall effect[15]. However, realizing such exotic spin textures in other ferromagnets remains a challenge.

[1]Laboratory of Advanced Materials, Shanghai Key Lab of Molecular Catalysis and Innovative Materials, Academy for Engineering & Technology, Fudan University, Shanghai 200438, PR China. [2]Shanghai Frontiers Science Research Base of Intelligent Optoelectronics and Perception, Institute of Optoelectronics, Fudan University, Shanghai 200433, PR China. [3]Materials Genome Institute, Shanghai University, Shanghai 200444, China. [4]Zhejiang Laboratory, Hangzhou 311100, China. [5]Pico Electron Microscopy Center, Innovation Institute for Ocean Materials Characterization, Center for Advanced Studies in Precision Instruments, Hainan University, Haikou 570228, China. [6]These authors contributed equally: Xiaowei Lv, Hualiang Lv, Yalei Huang. ✉e-mail: guixincao@shu.edu.cn; rcche@fudan.edu.cn

Moreover, the coexistence of multiple skyrmion phases within one material offers an additional degree of freedom for designing devices with improved properties in topology-based spintronics. For instance, a stream of binary data bits in magnetic memories can be encoded by a sequence of two distinct skyrmion phases[16]. This approach, as suggested by modeling work, has the potential to reduce error rates in racetrack memories where skyrmions serve as information carriers[17,18]. Furthermore, two-dimensional van der Waals magnets[19,20] have recently become attractive platforms for topology-based spintronics, owing to their exhibiting fascinating physical properties, such as giant tunneling magnetoresistance, strong spin-orbit coupling, and unconventional responses from quantum metamaterials. Magnetic skyrmions have also been successfully created in various 2D systems, such as the $Cr_2GeTe_6$[21], $Fe_3GeTe_2$[22–24], and $Fe_5GeTe_2$ family[25–27]. However, most of these materials possess a low Curie temperature ($T_c$) below the room temperature, restricting their further research and development in spintronic devices. More critically, to date, the coexistence of multiple soliton phases has only been reported in $Fe_5GeTe_2$ at 100 K[27]. Therefore, the quest for 2D materials hosting multiple topological phases above room temperature continues to be fueled by the promise of novel devices.

In this work, we reported the room-temperature observation of two distinct skyrmion phases, hybrid, and Bloch, in a 2D vdW magnet FGT by using Lorentz transmission electron microscopy (LTEM). Through systematic fitting of the spin textures' intensity profiles as a function of sample tilt under identical imaging conditions with LTEM simulations, the spin configurations of skyrmions and domain walls were identified well. The results reveal that the interplay between the dipolar interaction and DMI contributed to the stabilization of mixed Bloch-Neel topological spin textures with high thermostability.

## Results

### Identification of spin textures
Figure 1a–c illustrates the topological spin textures of the three types of skyrmions: Bloch, Neel, and hybrid skyrmions. The corresponding

helimagnetic structures from which these skyrmionic textures emerge are schematically depicted in the bottom row. Bloch skyrmions, typically stabilized in Chiral magnets and centrosymmetric cystals[1,2,28], exhibit a helicoid spin arrangement, where spins rotate in a plane perpendicular to the direction of propagation. Such skyrmions manifest as ring-like patterns when imaged by Lorentz transmission electron microscopy (LTEM) (see the second column of Fig. 1d–f). Crucially, there is no visible change in the LTEM contrast of Bloch skyrmions when the sample is tilted at positive or negative angles. As for Neel skyrmions, their cross-section shows cycloid spin propagation. At normal incidence, Neel skyrmions exhibit no contrast in LTEM images. And tilting the sample is essential to produce magnetic contrast, resulting in a half-bright and half-dark spot, which would be reversed at opposite tilt angles, as shown in the third column of Fig. 1d–f. The hybrid skyrmion, as its name implies, presents an intermediate state between helicoid and cycloid spin propagations. such mixed spin rotation is expected to produce a weaker ring-like pattern at zero tilt compared to Bloch skyrmions, and a half-bright and half-dark contrast similar to Neel skyrmions upon sample tilting, as presented by the fourth column of Fig. 1d–f. Therefore, it is expected that distinct skyrmion phases can be recognized by the analysis of tilt-dependent imaging contrast.

Single crystals of FGT were grown by the flux zone growth method[29]. It has a layered crystal structure, in which each $Fe_3Ga$ heterometallic slab is sandwiched by two Te layers along the c-axis. Figure 1g shows the High-angle annular dark-field-based STEM (HAADF-STEM) images along the [100] (left panel) and [001] (right panel) directions, displaying the atomic-scale structure and high quality of single-crystalline FGT magnet in this work. The interlayer spacing of 0.88 nm in FGT is slightly smaller than that in $Fe_5GeTe_2$ (0.98 nm), on account of lower Fe content[30]. This is further verified by the Energy-dispersive x-ray spectroscopy map, which shows the atomic-scale elemental distribution of Fe, Ga, Te, and atomic ratio of 3.35:1:2.11 (see Fig. S1 in SI). Subsequently, to investigate the magnetic properties of

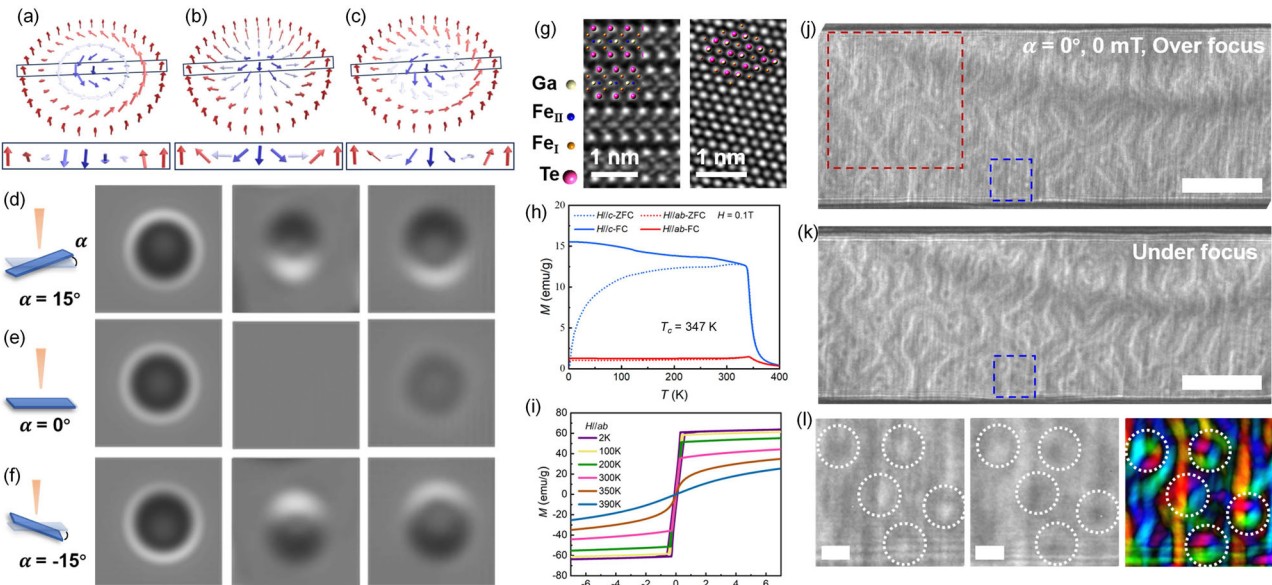

**Fig. 1 | Schematics of Skyrmions and characterization of FGT bulk crystal.**
**a–c** The upper panel shows the spin textures of a Bloch skyrmion (**a**), a Neel skyrmion (**b**), and a hybrid skyrmion (**c**), and the lower panel represents the corresponding cross-sections along the radial directions indicated by the solid rectangles. **d–f** The first column displays the schematic graphs of sample tilting for LTEM imaging. The second, third, and fourth columns represent the simulated LTEM pattern of Bloch, Neel, and hybrid skyrmions at different tilted angles, respectively. **g** Atomic-resolution HAADF-STEM images along the [100] (left panel) and [001] (right panel)

directions. **h** Temperature-dependent magnetization (M-T) curves measured from 2 to 400 K using the ZFC and FC protocols with a magnetic field of 1000 Oe. **i** Magnetic field dependence of magnetization (M-H) curves at temperatures varying from 2 to 390 K for H//c-axis. **j, k** Overfocused (Δf = +1 mm) (**j**) and underfocused (Δf = −1 mm) (**k**) LTEM images obtained at α = 0° and zero field at 290 K. The scale bars are 1 μm. **l** The enlarged view of the blue dotted box in (**j, k**) and in-plane magnetization map was obtained by using the TIE. The color wheel (inset) shows the direction and magnitude of magnetization. The scale bars are 100 nm.

the bulk FGT, temperature-dependent magnetization curves (M-T) were recorded using field-cooled (FC) and zero-field-cooled (ZFC) with a magnetic field of 0.1 T applied along the $c$-axis and $ab$-plane, as shown in Fig. 1h. The magnetization along the $c$-axis is significantly larger than in the $ab$-plane, similar with other van der Waals crystals such as $Fe_3GeTe_2$[22] and $Fe_5GeTe_2$[30], implying a predominantly out-of-plane magnetization. Crucially, the $T_c$ was identified to be 347 K, as determined by modified Curies-Weiss fitting (see Fig. S2 in SI). Such high $T_c$ makes it superior to most vdW magnets in futural 2D spintronic devices. Additionally, isothermal magnetization measurements $M(H)$ with a magnetic field parallel to the $c$-axis and $ab$-plane at various temperatures are plotted in Fig. 1i and Fig. S3 in SI, respectively. The smaller saturation field along the $c$-axis compared to the $ab$-plane indicates a strong perpendicular magnetic anisotropy in $Fe_3GaTe_2$, even at room temperature.

As magnetic skyrmions are usually stabilized in the magnetic crystals with out-of-plane anisotropy[28,31], thus to investigate the potential skyrmion-like textures in FGT, a 101-nm thick lamella perpendicular to the $c$-axis was prepared by using focused ion-beam milling (FIB). The thickness was estimated by the cross-section image using scanning electron microscopy (SEM) (see Fig. S4 in SI). To reduce thermal disturbance near the $T_c$, the lamella was initially zero-field-cooled (ZFC) from 347 to 290 K followed by LTEM imaging. The lamella was not tilted ($\alpha = 0°$) and thus the electron beam traveled along the $c$-axis of the crystal. Two electron micrographs of the zero-field state were obtained in over-focus and under-focus Fresnel modes ($\Delta f = \pm 1$ mm) of the LTEM, as shown in Fig. 1 j, k, respectively. Given that Neel walls do not show contrast at normal incidence, the magnetic contrast presented in the Fresnel images could be attributed to Bloch-type magnetization. One can observe that high density of skyrmion-like textures showing white and black dots existed as the magnetic ground states with the coexistence of stripes domain. Such magnetic equilibrium state is typically attributed to the competition between the perpendicular magnetic anisotropy, dipole-dipole interaction, and magnetic exchange interaction, etc[32,33]. By using the transport of intensity equation (TIE) based on the software QPT[34], the in-plane magnetization distribution of the blue-boxed region in Fig. 1j, k is presented in Fig. 1m. These white and black dots can be determined as clockwise and counterclockwise skyrmion textures, respectively. The presence of spontaneous zero-field skyrmions is attractive and potential in futural spintronic devices since there is no need for a magnetic field for the stabilization of skyrmions, which simplifies the systems and decreases the energy consumption of 2D topology-based memory devices. Besides, these textures resemble the dipolar skyrmions (Bloch type) in conventional centrosymmetric magnets, in which the skyrmions are stabilized by magnetic dipole-dipole interaction without DMI. However, they cannot be identified as Bloch skyrmions directly because the hybrid skyrmions with a mixed Bloch-Neel structure can produce the same contrast when the sample is not tilted. Furthermore, recent studies suggest the existence of weak DMI (-0.379 mJ·m$^{-2}$) in bulk FGT crystals, attributed to the asymmetric crystal structure caused by atomic vacancies[35]. This level of DMI in crystals is believed to be capable of altering the spin configurations in magnetic domain, as evidenced by the unconventional helicity polarization observed in $Fe_5GeTe_2$[25].

Therefore, to further explore the configurations of these textures in FGT, a series of over-focused LTEM images of red boxed region in Fig. 1j were obtained at different tilt angles ranging from −15° to 15°, as shown in Fig. 2a−e. We selected three different regions, in which a white and black dot as well as a domain wall were presented separately, and boxed them in blue, red, and yellow colors, respectively. Line scans of these three structures acquired from experimental raw data are shown in Fig. 2f−h, where the gray dotted lines represent the background intensity levels. One can observe that the radial imaging intensity of the white and black dots display opposing

Bloch components, as shown in Fig. 2f, g. Notably, their contrast profiles show no visible changes at each tilt angle, agreed well with the features of Bloch-type skyrmions, thereby indicating the centrosymmetric crystal of FGT. However, the imaging intensity of a domain wall under varying tilt conditions shows distinct behavior, as shown in Fig. 2h. At 0° tilt, the profile displays a combination of a peak and a valley, demonstrating the pure Bloch contribution. As the tilt angle increases, an asymmetric shift in the line scans becomes apparent. Particularly, the contrast was almost reversed at opposite angles (i.e., ±10° or ±15°), a hallmark of Neel-type domain walls. These observations underscore the presence of a mixed-character domain wall, corroborating the existence of DMI in FGT. For such a hybrid structure, it was reported that the quantification of Bloch and Neel contributions can also be achieved through the analysis of the tilt-dependent contrast[36]. Intriguingly, such a hybrid wall is expected to produce no contrast at specific tilt angles due to the counteraction between Bloch and Neel contrast, as illustrated in Figs. S5 and 6. Besides, the theoretical intensity curve of defocused simulated LTEM images for Bloch-type skyrmions with dual chirality and a hybrid domain wall are in excellent agreement with the experimental data, as shown in Fig. 2i−k.

## Coexistence of Bloch and hybrid skyrmions

After demonstrating the coexistence of zero-field Bloch skyrmions and hybrid domain walls in FGT, questions arise regarding the stability of these skyrmions and the potential for creating hybrid skyrmions from stripe domains under magnetic fields. To address these queries, we explore the stability and evolution of these textures under varying perpendicular and tilted magnetic fields at 290 K (see Fig. 3). In the absence of a tilt and at zero perpendicular magnetic fields, the contrasts of the domain wall appear slightly weaker compared to that of the high-density Bloch skyrmions, on account of the invisible contrast produced by Neel components (Fig. 3a). As the magnetic field increases, a gradual decrease in skyrmion density is observed, eventually reaching zero at 183 mT (Fig. 3b−e). This monotonous trend (see Fig. 3f) differs from the behaviors typically seen in conventional magnets[26,37], where skyrmion density first increases and then decreases. Moreover, the hybrid domain walls evolved and annihilated with skyrmions synchronously as the magnetic field increased. As shown in Fig. 3d, narrower stripes domain coexisting with several skyrmions at 155 mT were observed, with the latter marked by white circles. Notably, no additional type of skyrmions (i.e. hybrid-type) was observed in the field-driven process when the lamella was not tilted. Reasonably, the contrast of hybrid skyrmions under such intense magnetic fields likely becomes too weaker for detection, and an in-plane magnetic field component may be more conducive to generating skyrmions with Neel contributions[38]. As anticipated, when the lamella was tilted at 15° (Fig. 3g−k), a different type of skyrmion-like texture presenting half-dark and half-light contrast was observed with the coexistence of Bloch skyrmions at 102 mT, as boxed by red circles in Fig. 3j. This pattern aligns with the characteristics of Neel contributions and suggests the identification of these textures as hybrid skyrmions (discuss latter). Besides, the relatively low density of hybrid skyrmions, akin to that of conventional Neel skyrmions under a tilted magnetic field, can be attributed to the competition between Zeeman energy, Heisenberg exchange energy, uniaxial anisotropy energy, demagnetization energy, and DMI energy. As the magnetic field increased further, both Bloch skyrmions and hybrid skyrmions annihilated and came into being a ferromagnetic state eventually (Fig. 3l).

For the validation of observed hybrid skyrmions, the analysis of Lorentz contrast under various tilted conditions has proven to be effective and essential. However, it is necessary to mention that these textures require a fixed magnetic field for stability, and different tilted conditions may alter the magnetic field applied to the sample, potentially disrupting the spin configurations of the structures.

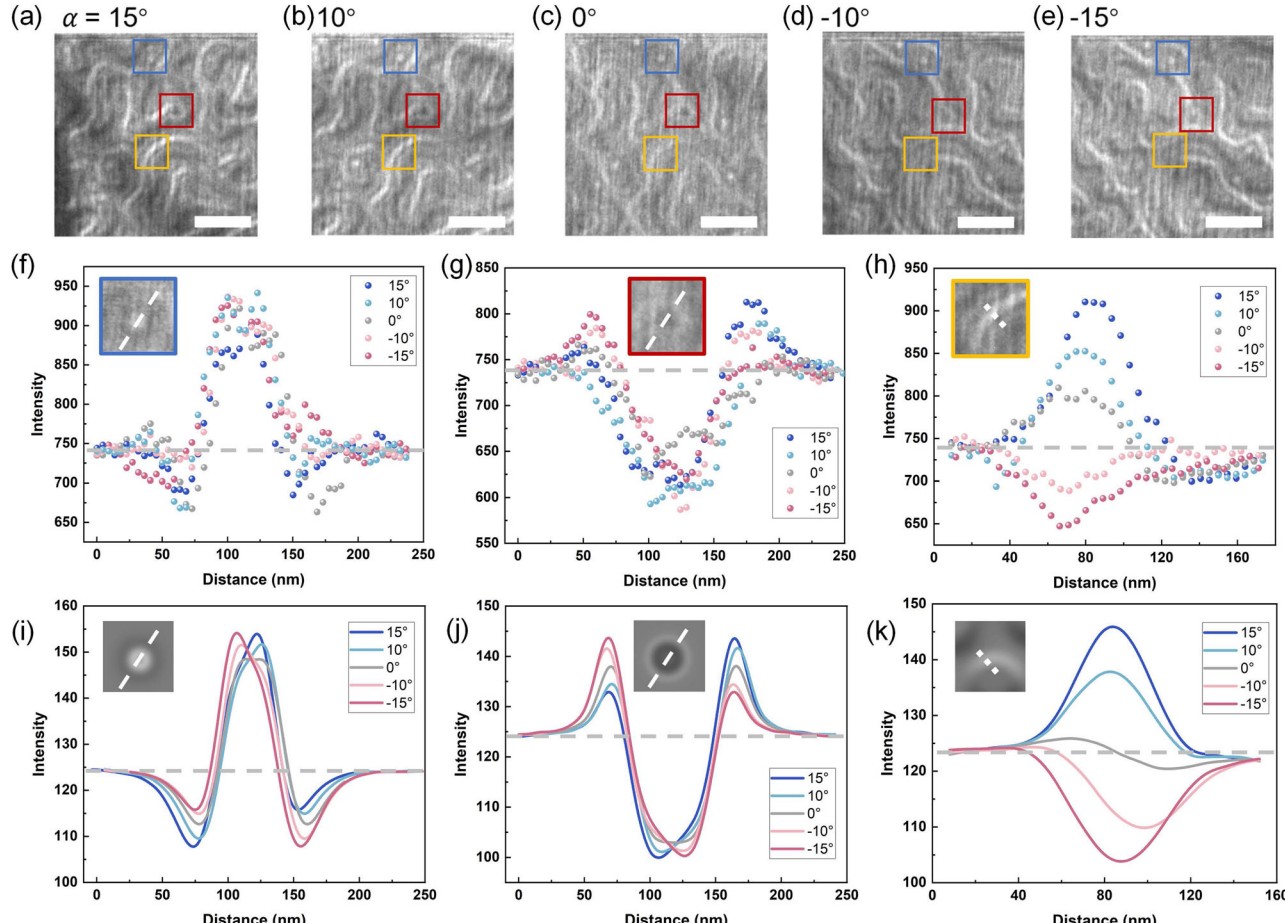

**Fig. 2 | The analysis of Bloch skyrmions and hybrid domain walls. a–e** Over-focused LTEM images acquired at −15° (**a**), 10° (**b**), 0° (**c**), −10° (**d**), and −15° (**e**) at 290 K. The same areas for analysis of three spin textures at tilted are boxed in blue, red, and yellow, respectively. **f–h** Experimental line scans from the red box (the inset of **f**), blue box (the inset of **g**), and yellow box (the inset of **h**) at different tilt angles. **i–k** Theoretical line scans from a Bloch skyrmion (the inset of **i**), a Bloch skyrmion with opposite helicity (the inset of **j**), and a hybrid domain wall (the inset of **k**) at different tilt angles. The intensity of the background in (**f–k**) is indicated by a gray dotted line.

Consequently, a zero-field condition might be more suitable for tilt-testing. Moreover, field-cooling FC_ protocol has been reported as a universal effective method to create high-density zero-field meta-stable skyrmion lattice, and the generation of spin textures is closely related to the magnetization history[23]. Therefore, a FC protocol was adopted, as schematically illustrated in Fig. S7. The FGT sample was cooled from 380 K (well above $T_c$) to 290 K with an out-of-plane magnetic field of 38 mT. It is necessary to mention that neither too small nor large magnetic field in field-cooling process is conducive to produce hybrid skyrmions, as presented in Fig. S8. After the FC operation, the magnetic field was turned off. Figure 4a displays three over-focused LTEM images obtained at 17°, 0°and −16° tilted at zero magnetic field, respectively. Starting with the middle panel of Fig. 4a, one can observe that high-density clockwise and counterclockwise Bloch skyrmions persist after the 38-mT FC process, as boxed and marked 1 and 2, respectively. Notably, another type of skyrmion-like textures with a larger size and weaker contrast was created, coexisting with Bloch skyrmions. As shown in the region labeled 3a–c of Fig. 4a, the magnetic contrast of these structures became more pronounced at non-zero tilt angles, exhibiting up-bright, down-dark, and up-dark, down-bright contrast at 17° and −16° tilt, respectively. These LTEM images are in excellent agreement with the theoretical images in Fig. 1d–f, demonstrating the spin configuration of hybrid-type sky-rmions. The representative images of two-chirality Bloch skyrmions and hybrid skyrmions in the same regions of each tilted image are

magnified in the left panels of Fig. 4b, and TIE was further used to analyze the in-plane magnetization distributions (right panels of Fig. 4b). It is important to note that while TIE is suitable for mapping the magnetization distribution of Bloch skyrmions, as their LTEM contrast remains unchanged upon tilting, it is less applicable for hybrid skyrmions due to the significant contrast variation at different tilt angles. Consequently, TIE results for hybrid textures at varying tilt angles exhibit distinct spin configurations, as illustrated in the boxed regions of Fig. 4b. Furthermore, heating experiments demonstrated that both coexisting phases can endure above room temperature, up to 328 K, highlighting their ultra-high thermostability, which can be attributed to the strong intrinsic ferromagnetism and large perpen-dicular magnetic anisotropy in FGT (see Fig. 4c and Figs. S10 and 11). Importantly, such high-thermostability magnetic phases could also display robust topological Hall effect (THE) over a broad temperature, which is pivotal for their electric detection in topology-based mem-ories, as illustrated in Fig. S12.

Emerging materials with distinct skyrmion phases at room tem-perature and even at zero fields are highly desirable for future spin-tronic applications. Figure 4d gives a comparison of FGT and other representative 2D material systems, including $CrCl_3$[39,40], $Fe_3GeTe$[22–24,41,42], $Fe_5GeTe2$[30,43], and $Cr_{0.87}Te$ family[44], in terms of topological texture phases and phase temperature at zero magnetic field after ZFC and FC process. Notably, after the FC process, all the candidates can stabilize at least one texture phase (either Neel- or

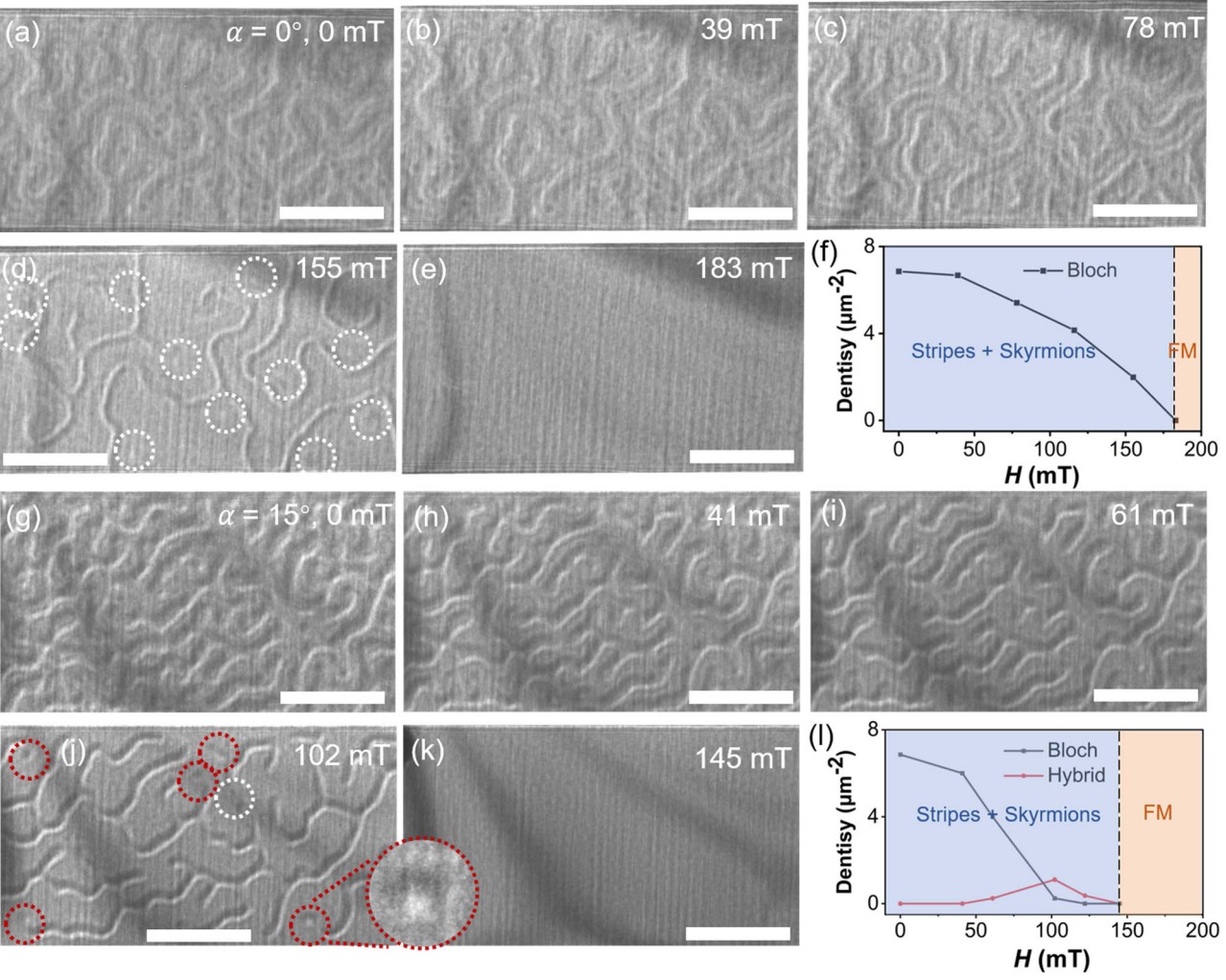

**Fig. 3 | Evolution of spin textures in FGT at 290 K. a–e** Over-focused LTEM images of Bloch skyrmions and hybrid stripes collected at different magnetic fields when the lamella is not tilted. **g–k** Over-focused LTEM images of Bloch skyrmions and hybrid stripes collected at different magnetic fields when the lamella is 15° tilted. White and red circles represent the Bloch skyrmions and hybrid skyrmions, respectively. The scale bars are 1 μm. The skyrmion density as a function of the magnetic field at 0° (**f**) and 15° (**l**) tilt.

Bloch-type skyrmions) at zero field. However, a common limitation among most of these materials is that the emergence of zero-field skyrmions occurs well below room temperature. In comparison, FGT stands out by hosting two distinct room-temperature skyrmion phases, namely Bloch and hybrid skyrmions with exceptional high thermostability. Additionally, skyrmions created through the FC strategy represent a metastable state, which is less advantageous compared to a stable state. Consequently, zero-field topological textures without the need for a FC process are more desirable. As demonstrated in the right panel of Fig. 4d, the spontaneous spin textures are typically limited to merons and are stabilized in materials with a low $T_c$. Fortunately, FGT can support spontaneous Bloch skyrmions at high densities and, importantly, above room temperature. These distinctive properties of FGT not only give it an edge over other known vdW materials but also potentially open avenues for room-temperature 2D memory devices incorporating distinct topological spin textures

**Micromagnetic simulations**
Generally, the LTEM provides a 2D real-space imaging that averaging the magnetic field over depth, which means that the characterizations of the exact 3D configurations of textures are challenging. Nevertheless, recent observations of skyrmion tubes[45], bobbers[16], and braids[46] in B20 chiral magnets as well as target skyrmion bubbles[47] in centrosymmetric crystals have demonstrated that skyrmions are complex three-dimensional (3D) string-like objects, which could conceptually extend through the thickness of a single crystal. Therefore, to gain a deeper understanding of the mixed skyrmion phases observed in the FGT system, 3D micromagnetic simulations were performed as a function of DMI to explore their depth-modulated spin configurations (see methods for the parameters used), as shown in Fig. 5. At $D = 0.0$ mJ/m², the skyrmion tube represents the Bloch character at $Z$ of 50 nm and mixed Bloch-Neel contributions at $Z$ of 0 and 100 nm, which can be attributed to the dipole-dipole interaction (see Fig. 5a). This finding aligns with the Neel-twisted skyrmionic configurations observed in near-surface layers of other centrosymmetric magnets[48]. More importantly, the surface layers host radially inverse inward- and outward-pointing Neel spins in the upper and bottom layers, respectively. This arrangement results in the average Neel magnetization canceling itself out, leading to the manifestation of pure Bloch skyrmions in LTEM imaging. As the $D$ value increases, more Neel components are generated in the skyrmion tube, on account of the competition between DMI and dipolar interaction (see Fig. S13 for details). As shown in Fig. 5b, the outward-pointing Neel spins represent in the both middle and bottom layers, while Bloch magnetization occurs in the upper layer at $D = 0.6$ mJ/m². This combination results in

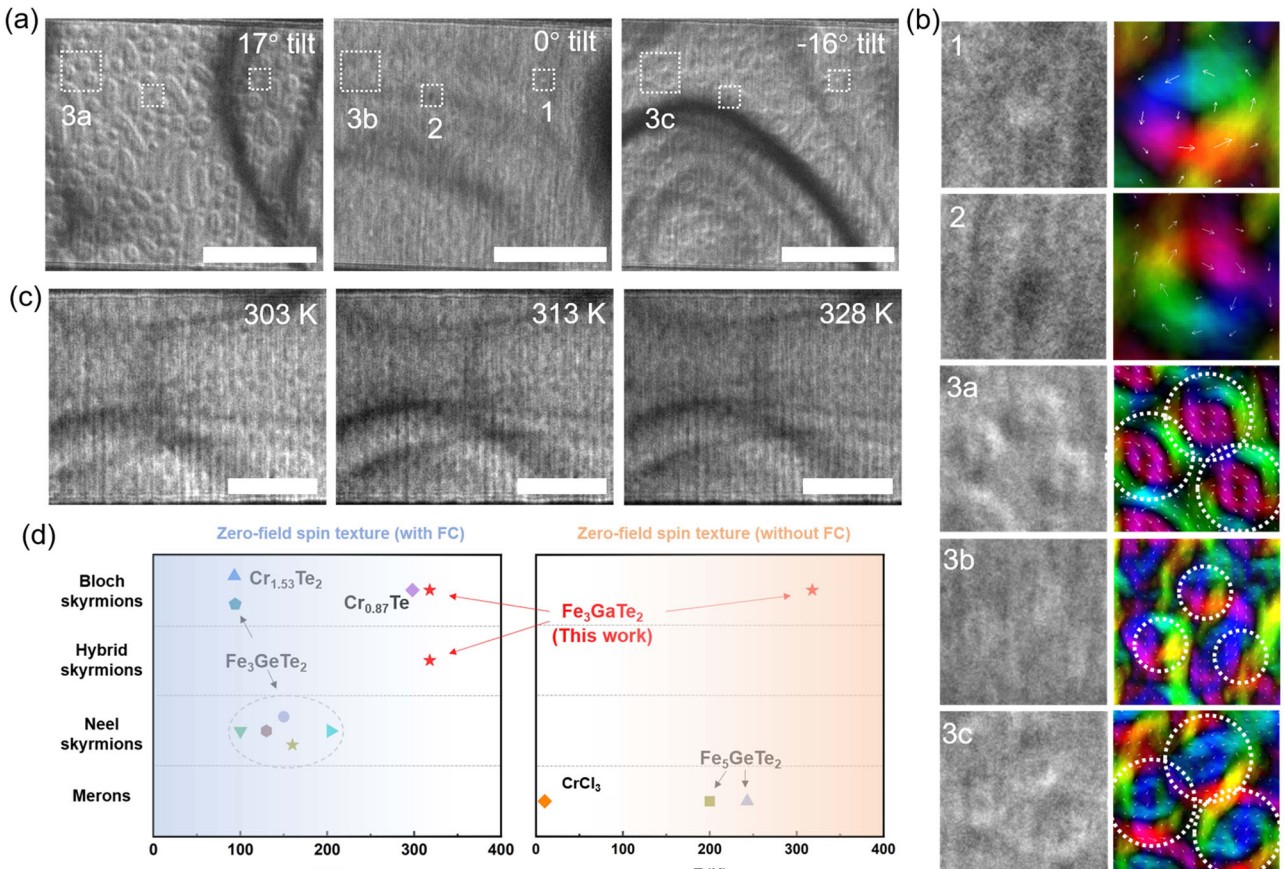

**Fig. 4 | The coexistence of above room-temperature Bloch skyrmions and hybrid skyrmions after the FC process. a** Over-focused LTEM images acquired at different tilt angles at 290 K after the FC process. Two helicity of Bloch skyrmions are boxed and marked by 1 and 2, respectively, and hybrid skyrmions in the same region are boxed and marked by 3a, 3b, and 3c at different tilt angles. **b** Left column: enlarged images in the boxed regions of a. Right column: the corresponding in-plane magnetization maps. **c** Over-focused LTEM images acquired at different temperatures at 0° tilt. The scale bars are 1 μm. **d** Comparison of topological textures phase and phase temperature in FGT and other 2D material systems. Expect for FGT, the other data are taken from refs. 22–24,30,39–44.

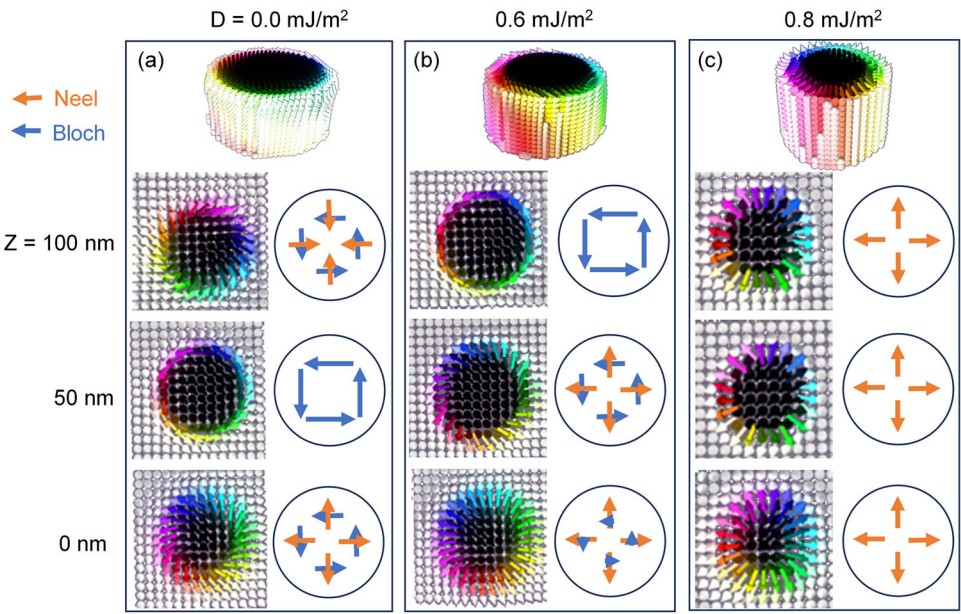

**Fig. 5 | Micromagnetic simulations. a–c** Visualizations of 3D spin configurations showing the magnetic transformations from Bloch- (**a**) to hybrid- (**b**) to Neel-type skyrmions (**c**) with increasing DMI strength. The upper row represents the side-view of skyrmion tubes, and the bottom three rows are average magnetic configurations at different heights Z, where the right panels show the schematic drawings of the left simulations. The orange and blue arrows denote the Neel and Bloch contributions of magnetization, respectively.

an averaged mixed Bloch-Neel magnetization, indicative of hybrid skyrmions. As the value of $D$ increases to 0.8 mJ/m², only Neel characters show in all layers of skyrmion tubes, forming the pure Neel skyrmions and demonstrating that the DMI dominates over magnetic dipole-dipole interaction. The truth is agreed well with the observation of Neel skyrmions in other DMI-prevailed systems[38]. Besides, the particular value of $D$ required to transform the spin configurations of skyrmions will depend on factors such as the values of the other micromagnetic parameter, and sample thickness. Therefore, it can be concluded that the value of $D$ of FGT lamella falls within the range of 0–0.6 mJ/m² by the observation of Bloch and hybrid skyrmions in this work. This value is comparable to the reported value of 0.379 mJ/m² in the pristine sample and significantly lower than the values of 7.354 or 2.301 mJ/m² in FGT with surface oxidation[35]. Crucially, the parameter $D$ estimated by simulations also aligns within the range of DFT-calculated results (see Figs. S14 and 15 in SI for details), emphasizing the pivotal role of deviation in $Fe_{II}$ atoms. This could elucidate the physical origin of DMI in FGT[49,50], thereby encouraging further exploration in this area.

## Discussion

We have successfully observed two distinct skyrmion phases at room temperature in 2D magnet FGT: spontaneously formed Bloch skyrmions constituting the magnetic ground state, alongside hybrid stripe domains capable of evolving into hybrid skyrmions under an external magnetic field. A combination of the real-space image and simulations suggests that the competition between DMI and dipole-dipole interaction is the primary stabilization mechanism of the mixed skyrmion phases. These objects represent a promising area for further fundamental research and possible practical applications. Hybrid skyrmions could potentially eliminate the skyrmion Hall effect in future studies. Besides, zero-field Bloch skyrmions may be attractive and potential in futural spintronic devices, since there is no need of a magnetic field for the stabilization of skyrmions, which simplifies the systems and decreases the energy consumption of 2D topology-based memory devices. Furthermore, the coexistence of these two skyrmion phases can survive even at 328 K, demonstrating their high thermostability. These findings suggest an alternative approach for data encoding in 2D spintronic devices, in which a data stream can be encoded in a single chain composed of two distinct particles at room temperature.

## Methods

### Magnetic and crystal characterization

The magnetization properties of the bulk FGT crystals were evaluated utilizing a SQUID-VSM system (MPMS, Quantum Design) across a temperature range of 2−400 K. Analyses of the microstructure, morphology, and molar ratio were conducted using an FEI Titan[3] Themis 60-300 instrument.

### LTEM observation and image simulation

For the LTEM observation, several lamellae from the same single crystal of FGT were prepared by Focused Ion Beam (FIB) Ga⁺ ion milling [Helios Nanolab 600I; FEI] using standard lift-out procedures. Magnetic domain structures were investigated using a modified JEM-2100F LTEM, equipped with a liquid-nitrogen holder (90–383 K), at an acceleration voltage of 200 kV. The high-resolution in-plane magnetic induction maps were processed by QPT software based on the TIE. The LTEM images were simulated by using the MALTS code.

### Micromagnetic simulations

Micromagnetic simulations based on the LLG function were performed using Mumax3[51]. We consider the Hamiltonian exchange interaction energy, magnetic anisotropy energy, Zeeman energy, dipole-dipole interaction energy, and varied DMI energy. Simulated magnetic parameters are set based on the bulk FGT crystal at room temperature: uniaxial anisotropy constant $K_u = 0.8 \times 10^5$ J·m⁻³, saturation magnetization $M_s = 2.4 \times 10^5$ A·m⁻¹, exchange stiffness constant $A = 5.0$ pJ·m⁻¹, Gibert damping constant $\alpha = 0.5$. The DMI was varied in the range of 0−0.8 mJ·m⁻². The simulation cell is set to $5 \times 5 \times 5$ nm³, while the system geometry is defined as a cuboid with 1000-nm length and 100-nm thickness.

## Data availability

The data that support the findings of this study are available from the corresponding author upon reasonable request.

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

## Acknowledgements

This work was supported by the National Natural Science Foundation of China (52231007, 12327804, T2321003, 22088101, 52301236), the Ministry of Science and Technology of China (973 Project No. 2021YFA1200600).

## Author contributions

X.L., H.L., and Y.H. contributed equally to this work. X.L. and R.C. conceived the idea and designed the experiments. Y.H., Y.D., and G.C. synthesized the sample and did the basic characterizations; X.L., M.L., and K.P. operated the LTEM and HAADF-STEM experiments under the guidance of R.C., J.Z., and Y. L.; X.L and K.P. preformed the micro-magnetic simulations; H.L., R.Z., and G.Q. carried out the first-principles calculations; X.L. and R.C. analyzed data and wrote the manuscript. All authors contributed to the discussion of the results and the improvement of the manuscript.

## Competing interests

The authors declare no competing interests.
