## [Peer Review File · Nature Communications]

Distinct Skyrmion Phases at Room Temperature in Two-dimensional Ferromagnet Fe₃GaTe₂REVIEWER COMMENTS

Reviewer #1 (Remarks to the Author):

The manuscript entitled "Observation of Distinct Skyrmion Phases at Room Temperature in 2D Ferromagnet Fe₃GaTe₂" by the authors of X. Lv et al. presents significant findings on the identification of multiple topological phases in a two-dimensional magnet Fe₃GaTe₂ at room temperature. The combination of Lorentz TEM (L-TEM) measurements and micromagnetic simulations to uncover the mixed Bloch-Neel character of magnetic domains in Fe₃GaTe₂ is commendable. Importantly, this study introduces an unprecedented type of topological structures, termed hybrid skyrmions, and demonstrates their coexistence with spontaneously formed Bloch skyrmions, exhibiting remarkable thermal stability even above room temperature. I find the present findings novel and interesting, and the experimental work is of good quality. The implications of this research for the development of room-temperature 2D spintronic devices utilizing skyrmions are substantial. Thus, I recommend this manuscript to be published in Nature Communications as a VIP article if the authors address the following questions in revision.

1. The manuscript states the presence of Dzyaloshinskii-Moriya interaction (DMI) in the FGT lamella, crucial for the mixed Bloch-Neel spin textures. However, this seems inconsistent with FGT's centrosymmetric crystal structure. Could the authors clarify the origin of DMI in this context? Is it related to the natural oxidation at the O-FGT/FGT interface, or are there other factors at play? Additional evidence to support this claim would greatly strengthen your argument.
2. On page 5, line 19, the statement "the lamella was initially zero-field cooled (ZFC) from 347 to 290 K" is confusing. It may raise doubts about whether the presence of Bloch skyrmions and hybrid stripes domains must go through the ZFC process, or if these structures are naturally present at room temperature without requiring ZFC. Could the authors provide clarification on this matter?
3. There seems to be a discrepancy between the appearance of hybrid skyrmions in Fig. 3j and those in Fig. 4 (3a-3c). Could you explain the differences observed? Additionally, the presence of stripe domains in FGT after the field-cooled (FC) process, as shown in Fig. 4a, warrants further discussion. How does the FC process affect these magnetic domains?
4. Regarding the simulation part, it is obvious that all skyrmion tubes exhibit varying degrees of hybridization between Bloch and Neel for different D values. In other words, can the Bloch skyrmions with $D = 0.0$ mJ/m² be identified as hybrid skyrmions to a certain degree as well?
5. Is there any signature of the transport properties associated with the topological spin texture? It is well known that skyrmions cause the topological Hall effect.

Below are a few minor points:

1. There are typos. The English should be checked.
2. The pictures should be carefully optimized, e.g., Fig. 3.
3. Page 4, Line 18 and Page 12, Line 7.
4. In Fig. 4d, there is a mistake regarding the reference, i.e., CrI₃.
5. Ensure consistency in the statement about T_c between Fig. 1i and Page 6 (Line 7)

Reviewer #2 (Remarks to the Author):

In this manuscript, the authors reported the experimental observation of mixed-type skyrmions in a room-temperature ferromagnet Fe₃GaTe₂ (FGT). They considered that the mixed-type skyrmions with both Bloch and hybrid characteristics exhibited high thermostability up to 328 K. This is an interesting finding and may be useful for the further experimental exploration of distinct skyrmion phases in 2D ferromagnets. However, there are some key issues that need to be further clarified. My detailed comments are given below.

1. The experimental results in Fig.(f-h) exhibit a visible deviation from the fitting results depicted in Fig.(i-k), particularly evident in the intensity peaks ranging from 150 nm to 200 nm. In my opinions, it is recommended to appropriately incorporate additional data point into Fig.2(f-g) to ensure consistency with the fitting outcomes in Fig.2(i-j).
2. The bulk crystalline Fe₃GeTe₂ magnet has been reported to exhibit a high Curie temperature of ~230 K (Nature 2018, 563, 94-99). Additionally, a single skyrmion bubble at ~120 K can be induced in Fe₃GeTe₂ (Nano Lett. 2020, 20, 868-873). What is inherent mechanism for inducing these significant differences between Fe₃GaTe₂ and Fe₃GeTe₂?
3. Dzyaloshinsky-Moriya interaction (DMI) is commonly observed in magnets with a space inversion symmetry breaking (Phys. Rev. B 2022, 106, 094403). However, the question remains: what is the underlying origin of the DMI that is necessary for generating hybrid skyrmion in this centrosymmetric Fe₃GaTe₂ magnet?
4. In previous laws, the number of skyrmions generally increases, then decreases, and eventually disappears when a perpendicular magnetic field is applied (Appl. Phys. Lett. 2022, 121, 202402). However, it is intriguing to note that in Figure 3(a-f), the addition of a vertical magnetic field leads to a gradual decrease in the number of skyrmions. Furthermore, in Figure S6, there is an initial increase followed by a subsequent decrease. Therefore, what could be the inner reason for this discrepancy between these two samples?
5. The micromagnetic simulations are widely used to probe the topological magnetic texture of magnetic materials. In Fig.3(a-f), a phase diagram should be provided to explore the potential trend of the skyrmion number of the two samples and determine an boundary among the different magnetic phases.
6. The observation of both Bloch and hybrid skyrmions is interesting. However, what cause the mixing of a skyrmion with two compounds? Could it be attributed to crystal defects or impurity adsorption in the material?
7. The experimental results and micromagnetic simulation results exhibit excellent agreement, indicating a strong correlation between the data. How are the parameters of micromagnetic simulations derived from the experiments, and what is their underlying basis? If possible, the authors can make some relevant first-principle calculations.
8. Can the authors provide magnetotransport measurements of 2D room-temperature ferromagnet Fe₃GaTe₂ to understand the transport characteristics of mixed-type skyrmions?
9. The scale bars of the graphs, such Fig.2(k,l), are not illustrated.

Dear Editor and Reviewers:

We really appreciate your letter and the reviewers' valuable comments about this paper, which have helped us tremendously to improve this revised manuscript. We have carefully considered the reviewers' comments and revised our manuscript. The main corrections in the paper and the point-by-point responses to the reviewers' comments are listed below.

Reviewer 1:

The manuscript entitled "Observation of Distinct Skyrmion Phases at Room Temperature in 2D Ferromagnet Fe₃GaTe₂" by the authors of X. Lv et al. presents significant findings on the identification of multiple topological phases in a two-dimensional magnet Fe₃GaTe₂ at room temperature. The combination of Lorentz TEM (L-TEM) measurements and micromagnetic simulations to uncover the mixed Bloch-Neel character of magnetic domains in Fe₃GaTe₂ is commendable. Importantly, this study introduces an unprecedented type of topological structures, termed hybrid skyrmions, and demonstrates their coexistence with spontaneously formed Bloch skyrmions, exhibiting remarkable thermal stability even above room temperature. I find the present findings novel and interesting, and the experimental work is of good quality. The implications of this research for the development of room-temperature 2D spintronic devices utilizing skyrmions are substantial. Thus, I recommend this manuscript to be published in Nature Communications as a VIP article if the authors address the following questions in revision.

Response: We sincerely thank the reviewer for careful reading of our manuscript and for recommending our manuscript to be published in Nature Communications. In the following we have thoroughly addressed all the comments raised by the reviewer.

1. The manuscript states the presence of Dzyaloshinskii-Moriya interaction (DMI) in the FGT lamella, crucial for the mixed Bloch-Neel spin textures. However, this seems inconsistent with FGT's centrosymmetric crystal structure. Could the authors clarify the

origin of DMI in this context? Is it related to the natural oxidation at the O-FGT/FGT interface, or are there other factors at play? Additional evidence to support this claim would greatly strengthen your argument.

Response: We sincerely thank the reviewer for the valuable comments. It is indeed that the presence of DMI is not consistent with Fe_3GaTe_2 's centrosymmetric crystal structure. Actually, Fe_3GaTe_2 has a similar crystal structure with Fe_3GeTe_2 , in which the Fe content plays a key role in many properties, including the Saturation magnetization (M_s), Curie temperature (T_c), crystal structure, space group, etc. As for the $\text{Fe}_{3+x}\text{GaTe}_2$ in this work, the Fe content is about 3.35 and the extra Fe atoms may contribute to the non-centrosymmetric crystals. Furthermore, recent studies claimed that the displacement deviation of Fe_{II} atom in FGT can induce the DMI [Nat. Commun. 15, 1017, (2024)] [Adv. Sci. 10, 2303443, (2023)], as shown in Fig. R1a. To confirm the hypothesis, we have further acquired an improved HAADF-STEM image, as shown in Fig. R1b-c. Subsequently, for a quantitative determination of the displacement of the Fe_{II} atom, we vertically integrated the corresponding imaging intensity line profile (Fig. R1d). By referencing the center of the two Te atoms, the deviation of the Fe_{II} atom towards the c axis was determined to be -0.09 \AA . Therefore, it is reasonable to believe that the FGT in this work is not an ideally centrosymmetric crystal structure.

To further investigated the relationship between the deviation of the Fe_{II} atom and DMI constant d , density functional theory (DFT) based first-principles calculations were employed, as shown in Fig. R2. Detailed about the calculations have been incorporated in revised supplementary information on page 12. It is evident that the absence of Fe_{II} atom deviation yields $D = 0 \text{ mJ/m}^2$, indicating an ideally centrosymmetric crystal structure of FGT, as shown in Fig. R2c. With the increment of Fe_{II} atom deviation, the value of D increases monotonously and reach to 0.94 mJ/m^2 at 0.1 \AA deviation. It should be noted that the estimated value of D ($< 0.6 \text{ mJ/m}^2$) obtained by micromagnetic simulations also falls within the range of DFT-calculated values, demonstrating the credibility of our results.

Indeed, a O-FGT/FGT interface could induce a interfacial DMI, which is capable of the formation of Neel-type skyrmions, as reported in a recent work (see reference 35).

However, the value of DMI arising from the O-FGT/FGT interface (7.354 or 2.301 mJ/m²) is significantly higher than that in this work (≤ 0.6 mJ/m²). Furthermore, each FGT lamella that prepared by focused ion beam (FIB) system were put into the lens cone of TEM without delay for the LTEM observation. Therefore, we propose that negligible surface oxidation did not affect the domain structure in our experiments.

In addition, we have added the following general description about the DMI calculations in the revised version (refer to Page 15, Lines 2-6 in the main text and Page 12-13 in the supplementary information) and Supplementary Fig. S14-15, as also shown below:

“Crucially, the parameter D estimated by simulations also aligns within the range of DFT-calculated results (see Fig. S14-S15 in SI for details), emphasizing the pivotal role of the deviation in Fe_{II} atoms. This could elucidate the physical origin of DMI in FGT,^{49,50} thereby encouraging further exploration in this area.”

“Recent studies have asserted that the displacement deviation of Fe_{II} atom in FGT can induce the DMI,^{2,3} as illustrated in Fig. S14a. To validate the hypothesis, we have further acquired an improved HAADF-STEM image, as depicted in Fig. S14b-c. Subsequently, for a quantitative determination of the displacement of the Fe_{II} atom, we performed a vertical integration of the corresponding imaging intensity line profile (Fig. S14d). By referencing the midpoint of the two Te atoms, the deviation of the Fe_{II} atom towards the c -axis was determined to be -0.09 Å.”

“To further investigated the relationship between the deviation of the Fe_{II} atom and DMI constant D , density functional theory (DFT)-based first-principles calculations were employed, as depicted in Fig. S15. The calculation of the DMI vector occurred in two steps. Initially, structural relaxations were performed with a fixed $\delta(\text{Fe})$ using Gaussian smearing until the forces diminished to less than 0.001 eV/Å. Subsequently, spin-orbit coupling was integrated into the calculation, and the system’s total energy was determined based on the spin configuration, as illustrated in Fig. S15b. The parameter d was determined as $(E_{\text{CCW}}-E_{\text{CW}})/12$, where E_{CCW} denotes the energy of the

counter-clockwise configuration and E_{CW} denotes the energy of the clockwise configuration.⁴ The DMI constant D was then derived using the equation $D = 3\sqrt{2}d/(N_F a^2)$, where N_F represents the number of atomic layers, a is the lattice constant, and d represents the DMI strength. In the second step, the EDIFF parameter was set to 10^{-6} eV, and the tetrahedron method was employed to obtain an accurate total energy. The results are depicted in Fig. S15c. It is clear that the absence of Fe_{II} atom deviation yields $D = 0$ mJ/m², indicating an ideally centrosymmetric crystal structure of FGT. As the Fe_{II} atom deviation increases, the value of D increases monotonously and reaches 0.94 mJ/m² at a 0.1 Å deviation. This value might be slightly larger than the actual one in our experiments, which is attributed to the omission of some realistic factors such as temperature and nonuniformity due to DFT's limitations. It should be highlighted that the estimated value of D (< 0.6 mJ/m²) obtained through micromagnetic simulations also aligns within the range of DFT-calculated values, underscoring the credibility of our findings.”

Fig. R1 The analysis of crystal structure of FGT. **a** Schematic of atom arrangements in centrosymmetric and asymmetric FGT along c axis. **b** Observation of Fe_{II} atom deviation. The region is selected from a atomic-resolution HAADF-STEM image (c). **d** Line profile of the image intensity shown in (b). The blue and red lines represent the central position of two Te atoms and Fe_{II} atom, and their distance is estimated at 0.09 \AA , respectively.

Fig. R2 The results of first-principles calculations. **a** Schematic of DMI in FGT. The orange arrow D_1 shows the direction of DMI vector induced by Fe_{II} and top Te atoms, while the green arrow D_2 represents the opposite direction of DMI vector induced by Fe_{II} and bottom Te atoms. D_1 is not equal to D_2 , leading to a net DMI D_{eff} . **b** Spin configurations of counter-clockwise (CCW) (left column) and clockwise (CW) (right column). **c** Fe_{II} atom deviation-dependent D obtained by first-principles calculations.

2. On page 5, line 19, the statement "the lamella was initially zero-field cooled (ZFC) from 347 to 290 K" is confusing. It may raise doubts about whether the presence of Bloch skyrmions and hybrid stripes domains must go through the ZFC process, or if these structures are naturally present at room temperature without requiring ZFC. Could the authors provide clarification on this matter?

Response: We thank the reviewer's comments on ZFC process. Actually, the presence of Bloch skyrmions and hybrid stripes is spontaneous at room temperature and zero magnetic field without requiring ZFC process. However, for Lorentz experiments, the FGT lamella is easy to be magnetized by the residual field of objective lens when the sample holder is inserted, though the objective lens is turned off. Therefore, to ensure

the observation of magnetic ground state, rather than the metastable state, we initially raised the sample temperature above Curie temperature (347 K), then cooled it to the target temperature (290 K).

3. There seems to be a discrepancy between the appearance of hybrid skyrmions in Fig. 3j and those in Fig. 4 (3a-3c). Could you explain the differences observed? Additionally, the presence of stripe domains in FGT after the field-cooled (FC) process, as shown in Fig. 4a, warrants further discussion. How does the FC process affect these magnetic domains?

Response: We sincerely thank the reviewer for the valuable comments. We agree with the reviewer that there is a discrepancy between the appearance of hybrid skyrmions in Fig. 3j and those in Fig. 4 (3a-3c). However, the difference is only about the size of hybrid skyrmions, rather than the skyrmions type. For a precise comparison of magnetic domain in Fig. 3j and Fig. 4a, several skyrmions are marked and enlarged, as shown in the Fig. R3. It is obvious that all skyrmions possess a half-dark and half-light contrast. More importantly, the skyrmion marked as 1 has a comparable size with the skyrmions marked as 2 and 3, all of which are smaller than the skyrmion marked as 4. The reason about different-sized skyrmions can be attributed to the competition between magnetic dipole-dipole interaction energy and DMI energy, as reported in recent studies (see [Nat. Commun. 15, 1017, (2024)] and [Adv. Mater. 2311022, (2024)]).

Following the reviewer's comments, we have supplemented the experiments about the FC process-dependent magnetic domain (Fig. R4) and some discussion in both the revised main text (refer to Page 11, Lines 6-8) and Supplementary information (refer to Page 6, Lines 10-17, as also shown below:

“It is necessary to mention that neither too small nor large magnetic field in field-cooling process is conducive to produce hybrid skyrmions, as presented in Fig. S8.”

“The results of FC-dependent magnetic domain are presented in Fig. S8. It is evident that only Bloch skyrmions and hybrid stripes were observed in FGT after a zero magnetic field cooled (ZFC), as shown in Fig. S8a. With the increment of magnetic field in FC process (see Fig. S8b-c), more hybrid skyrmions were observed and the

zero-field hybrid skyrmions lattice were created after the 512 mT-FC process. Notably, increasing the magnetic field further would be not conducive to the formation of skyrmions, and it came in being a ferromagnetic state nearly after the 896 mT-FC process, as presented in Fig. S8d.

Fig. R3 The comparison between the skyrmions under a magnetic field of 102 mT (a) and after the FC process (b). **c** Enlarged images of skyrmions in the boxed regions in (a) and (b).

Fig. R4 Magnetic skyrmions in FGT observed at 290 K and zero magnetic field after the FC process with a magnetic field of 0 mT (a), 256 mT (b), 512 mT (c) and 896 mT (d). The tilt angle is 0° (a), and 10° (b-d). The scale bar is $1 \mu\text{m}$.

4. Regarding the simulation part, it is obvious that all skyrmion tubes exhibit varying degrees of hybridization between Bloch and Neel for different D values. In other words, can the Bloch skyrmions with $D = 0.0 \text{ mJ/m}^2$ be identified as hybrid skyrmions to a certain degree as well?

Response: We agree with the reviewer that Bloch skyrmions with $D = 0.0 \text{ mJ/m}^2$ can be identified as hybrid skyrmions to a certain degree. In fact, all skyrmion tubes present different degrees of hybridization between Bloch and Neel for different D values, which can be attributed to the magnetic dipole-dipole interaction. Nevertheless, as for the Bloch skyrmions, the radially inverse inward- and outward-pointing Neel spins in the upper and bottom surface result in the negligible average Neel magnetization, which leads to the manifestation of pure Bloch skyrmions in LTEM imaging. It is identified as conventional dipolar skyrmion as well.

5. Is there any signature of the transport properties associated with the topological spin texture? It is well known that skyrmions cause the topological Hall effect.

Response: We appreciate the reviewer's insightful question. The topological Hall

effect (THE) is indeed considered as a hallmark for the identification of nonlinear spin textures. Therefore, to investigate the potential THE in FGT, we fabricated a Hall-bar device with a sample thickness of about 19.58 μm for the electric transport measurements over the temperature range of 300 -100 K, as shown in Fig R5. The results show that there is a discrepancy at the low-field region in magnetic hysteresis of Hall resistivity ρ_{xy} , demonstrating the presence of a pronounced THE component (see Fig. R5b). By linearly fitting the ρ_{xy} , we obtained the field dependent extracted topological Hall resistivity ρ_{xy}^T at various temperature (see Fig. R5c-d). It is evident that the THE signals are remarkable at 100 K and the maximum value of ρ_{xy}^T reach to 1.46 $\mu\Omega$ cm. With the increment of temperature, the ρ_{xy}^T reduces monotonically but even apparent at 300 K with a maximum value of 0.73 $\mu\Omega$ cm. The existence of such broad-temperature THE signals shows good agreement with the observations of topological skyrmions by LTEM.

In the revised version, we have added the associated discussions on transport properties on Page 12, Lines 9-12 in the main text and Page 9, Lines 9-20 in the Supplementary information, as also shown below:

“Importantly, such high-thermostability magnetic phases could also display robust topological Hall effect (THE) over a broad temperature, which is pivotal for their electric detection in topology-based memories, as illustrated in Fig. S12.”

“To investigate the potential THE in FGT, we fabricated a Hall-bar device with a sample thickness of about 19.58 μm for the electric transport measurements over the temperature range of 300 -100 K, as shown in Fig. S12. Fig. S12a shows the schematic of the FGT Hall device with $I // ab$ plane and $B // c$ axis. The results show that there is a discrepancy at the low-field region in magnetic hysteresis of Hall resistivity ρ_{xy} , demonstrating the presence of a pronounced THE component (see Fig. S12b). By linearly fitting the ρ_{xy} , we obtained the field dependent extracted topological Hall resistivity ρ_{xy}^T at various temperature (see Fig. S12c-d). It is evident that the THE signals are remarkable at 100 K and the maximum value of ρ_{xy}^T reach to 1.46 $\mu\Omega$ cm. With the increment of temperature, the ρ_{xy}^T reduces monotonically but even apparent at

300 K with a maximum value of $0.73 \mu\Omega \text{ cm}$. The existence of such broad-temperature THE signals shows good agreement with the observations of high-thermostability topological skyrmions by LTEM.”

Fig. R5 Transport properties of FGT nanoflake. **a** Schematic of the FGT Hall device with $I \parallel ab$ plane and $B \parallel c$ axis. **b** Temperature-dependent Hall resistivity ρ_{xy} . **c** Representative extraction of topological Hall resistivity ρ_{xy}^T at 300 K. ρ_{xy}^N and ρ_{xy}^A shows the ordinary and anomalous Hall resistivity. **d** Temperature-dependent topological Hall resistivity ρ_{xy}^T . Blue and red curves were measured with decreasing and increasing magnetic field, respectively.

Below are a few minor points:

1. There are typos. The English should be checked.

Response: We thank the reviewer for catching these mistakes. The typos and English have now been corrected and improved in the revised manuscript.

2. The pictures should be carefully optimized, e.g., Fig. 3.

Response: We thank the reviewer for the valuable suggestion. We have carefully

optimized the Fig.3, as presented in the revised manuscript (page 23).

3. Page 4, Line 18 and Page 12, Line 7.

Response: We thank the reviewer for catching these mistakes. We have checked and corrected these mistakes in the revised version of the manuscript.

4. In Fig. 4d, there is a mistake regarding the reference, i.e., CrI3.

Response: Thanks for your comment. We have checked and corrected these mistakes in the revised version (page 24).

5. Ensure consistency in the statement about T_c between Fig. 1i and Page 6 (Line 7)

Response: Thanks for your comment. We have checked the statement about the T_c and corrected it in the revised manuscript (page 21).

Reviewer 2:

In this manuscript, the authors reported the experimental observation of mixed-type skyrmions in a room-temperature ferromagnet Fe₃GaTe₂ (FGT). They considered that the mixed-type skyrmions with both Bloch and hybrid characteristics exhibited high thermostability up to 328 K. This is an interesting finding and may be useful for the further experimental exploration of distinct skyrmion phases in 2D ferromagnets. However, there are some key issues that need to be further clarified. My detailed comments are given below.

Response: We sincerely thank the reviewer for careful reading of our manuscript and for pointing out that our finding is interesting and useful for the further exploration of skyrmions. The valuable suggestions and comments are greatly helpful to improve our manuscript. Below we answer the reviewer's questions and comments in a point-by-point basis. We hope the reviewer will be satisfied with the revised manuscript as well as our responses.

1. The experimental results in Fig.(f-h) exhibit a visible deviation from the fitting results depicted in Fig.(i-k), particularly evident in the intensity peaks ranging from 150 nm to 200 nm. In my opinions, it is recommended to appropriately incorporate additional data point into Fig.2(f-g) to ensure consistency with the fitting outcomes in Fig.2(i-j).

Response: We express our gratitude to the reviewer for their valuable suggestion. The suggestion gives us an important indication that the number of data points was not enough so as to make the deviation between the micromagnetic simulation fitting and experimental LTEM images looks a little bit obvious. Following this meaningful suggestion, we have incorporated additional data points from the LTEM images into Figures 2f and 2g in the revised manuscript to ensure consistency with the fitting outcomes depicted in Figures 2i and 2j, as illustrated in Fig. R6.

Fig. R6 The new version of Fig. 2 in the revised manuscript.

2. The bulk crystalline Fe_3GeTe_2 magnet has been reported to exhibit a high Curie temperature of ~ 230 K (Nature 2018, 563, 94-99). Additionally, a single skyrmion bubble at ~ 120 K can be induced in Fe_3GeTe_2 (Nano Lett. 2020, 20, 868-873). What is the inherent mechanism for inducing these significant differences between Fe_3GaTe_2 and Fe_3GeTe_2 ?

Response: We agree with the reviewer in identifying significant differences between Fe_3GaTe_2 and Fe_3GeTe_2 in terms of their Curie temperature (T_c), magnetic anisotropy (K_u), saturation magnetization (M_s), and the conditions for skyrmions formation, among others. To the best of our knowledge, these differences can be attributed to the distinctions in their crystal structure, electronic configuration, and the specific interactions occurring within these materials, as illustrated in Table R1. For instance, the magnetism is strongly dependent on the spacing between layers and the interactions between Fe, Ge (Ga) and Te atoms. The substitution of Ge by Ga in these compounds can significantly alter the crystal field environment and the magnetic exchange

interaction, leading to different magnetic behaviors. More importantly, the electronic configuration and the density of states near the Fermi level play crucial roles in determining the magnetic properties of materials. The presence of Ga instead of Ge can lead to a different band structure, affecting the magnetic exchange interactions between the Fe moments, as evidenced by [Nature 2018, 563, 94-99] and Ref. 29. These interactions are critical for the establishment of magnetic order and T_c . Besides, the nature of magnetic exchange interactions, such as direct exchange, super-exchange, and RKKY (Ruderman-Kittel-Kasuya-Yosida) interactions, can vary significantly between Fe_3GaTe_2 and Fe_3GeTe_2 due to the differences in their electronic structures and crystal geometries, as evidenced by [arXiv:2402.14618 (2024)]. These interactions determine the strength and nature of the magnetic ordering, contributing to the high Curie temperature in Fe_3GaTe_2 .

As for the stabilization of skyrmions, the key factors are the K_u and M_s of the magnets. It was reported that appropriately increasing M_s could promote the formation of magnetic bubbles, and too large or small K_u is destructive to the stability of bubbles, see Ref. 26 and [ACS Appl. Mater. Interfaces 11, 12098 (2019)]. Therefore, the discrepancies on the K_u and M_s between Fe_3GaTe_2 and Fe_3GeTe_2 attribute to the distinct skyrmions condition. Furthermore, the existence of DMI or not in systems and its magnitude would impact the formation of skyrmions, as evidenced by the observations of Neel- or Bloch-type skyrmions as well as the unconventional polarization in different FGT studies (see Ref. 25).

Table R1. Comparison of critical parameters between Fe_3GaTe_2 and Fe_3GeTe_2 .

	Lattice constant (a)	Lattice constant (c)	K_u (300K)	M_s (300K)	Electric conductivity ($10^4 \Omega^{-1} \text{cm}^{-1}$)	Inter-plane interactions ($\text{Fe}_1\text{-Fe}_2$, $\text{Fe}_1\text{-Fe}_3$)	In-plane exchanges ($\text{Fe}_1\text{-Fe}_1$, $\text{Fe}_3\text{-Fe}_3$)	Average magnetic moment for Fe atoms	Formation energy
Fe_3GeTe_2	3.991 Å	16.33 Å	0	0	0.67	FM	AFM	$1.61 \mu_B$	48.9 meV/atom
Fe_3GaTe_2	3.986 Å	16.22 Å	$0.8 \times 10^5 \text{J.m}^{-3}$	$2.4 \times 10^5 \text{A.m}^{-1}$	0.12	FM	FM	$1.80 \mu_B$	36.85 meV/atom

3. Dzyaloshinsky-Moriya interaction (DMI) is commonly observed in magnets with a

space inversion symmetry breaking (Phys. Rev. B 2022, 106, 094403). However, the question remains: what is the underlying origin of the DMI that is necessary for generating hybrid skyrmion in this centrosymmetric Fe₃GaTe₂ magnet?

Response: We sincerely thank the reviewer for the valuable comments. Recent studies claimed that the displacement deviation of Fe_{II} atom in FGT can induce the DMI [Nat. Commun. 15, 1017, (2024)] [Adv. Sci. 10, 2303443, (2023)], as illustrated in Fig. R7a. To confirm the hypothesis, we have further acquired an improved HAADF-STEM image, as shown in Fig. R7b-c. Subsequently, for a quantitative determination of the displacement of the Fe_{II} atom, we vertically integrated the corresponding imaging intensity line profile (Fig. R7d). By referencing the center of the two Te atoms, the deviation of the Fe_{II} atom towards the c axis was determined to be -0.09 \AA . Therefore, it is reasonable to believe that the FGT in this work is not an ideally centrosymmetric crystal structure.

To further investigate the relationship between the deviation of the Fe_{II} atom and DMI constant d , density functional theory (DFT) based first-principles calculations were employed, as shown in Fig. R8. Detailed about the calculations have been incorporated in revised supplementary information on page 12. It is evident that the absence of Fe_{II} atom deviation yields $D = 0 \text{ mJ/m}^2$, indicating an ideally centrosymmetric crystal structure of FGT, as shown in Fig. R8c. With the increment of Fe_{II} atom deviation, the value of D increases monotonously and reach to 0.94 mJ/m^2 at 0.1 \AA deviation. This value may be slightly larger than the actual one in our experiments, which is attributed to that some effect of realistic factors including the temperature and nonuniformity are not considered due to the limitation of DFT. It should be noted that the estimated value of D ($< 0.6 \text{ mJ/m}^2$) obtained by micromagnetic simulations also falls within the range of DFT-calculated values, demonstrating the credibility of our results.

Following the reviewer's comments, we have added the added the following general discussion about the DMI origin in the revised version (refer to Page 15, Lines 2-6 in the main text and Page 12-13 in the supplementary information) and Supplementary Fig. S14-15, as also shown below:

“Crucially, the parameter D estimated by simulations also aligns within the range of DFT-calculated results (see Fig. S14-S15 in SI for details), emphasizing the pivotal role of the deviation in Fe_{II} atoms. This could elucidate the physical origin of DMI in FGT,^{49,50} thereby encouraging further exploration in this area.”

“Recent studies have asserted that the displacement deviation of Fe_{II} atom in FGT can induce the DMI,^{2,3} as illustrated in Fig. S14a. To validate the hypothesis, we have further acquired an improved HAADF-STEM image, as depicted in Fig. S14b-c. Subsequently, for a quantitative determination of the displacement of the Fe_{II} atom, we performed a vertical integration of the corresponding imaging intensity line profile (Fig. S14d). By referencing the midpoint of the two Te atoms, the deviation of the Fe_{II} atom towards the c -axis was determined to be -0.09 \AA .”

“To further investigated the relationship between the deviation of the Fe_{II} atom and DMI constant D , density functional theory (DFT)-based first-principles calculations were employed, as depicted in Fig. S15. The calculation of the DMI vector occurred in two steps. Initially, structural relaxations were performed with a fixed $\delta(\text{Fe})$ using Gaussian smearing until the forces diminished to less than 0.001 eV/\AA . Subsequently, spin-orbit coupling was integrated into the calculation, and the system’s total energy was determined based on the spin configuration, as illustrated in Fig. S15b. The parameter d was determined as $(E_{\text{CCW}}-E_{\text{CW}})/12$, where E_{CCW} denotes the energy of the counter-clockwise configuration and E_{CW} denotes the energy of the clockwise configuration.⁴ The DMI constant D was then derived using the equation $D = 3\sqrt{2}d/(N_F a^2)$, where N_F represents the number of atomic layers, a is the lattice constant, and d represents the DMI strength. In the second step, the EDIFF parameter was set to 10^{-6} eV , and the tetrahedron method was employed to obtain an accurate total energy. The results are depicted in Fig. S15c. It is clear that the absence of Fe_{II} atom deviation yields $D = 0 \text{ mJ/m}^2$, indicating an ideally centrosymmetric crystal structure of FGT. As the Fe_{II} atom deviation increases, the value of D increases monotonously and reaches 0.94 mJ/m^2 at a 0.1 \AA deviation. This value might be slightly larger than the actual one

in our experiments, which is attributed to the omission of some realistic factors such as temperature and nonuniformity due to DFT's limitations. It should be highlighted that the estimated value of D (< 0.6 mJ/m²) obtained through micromagnetic simulations also aligns within the range of DFT-calculated values, underscoring the credibility of our findings.”

Fig. R7 The analysis of crystal structure of FGT. **a** Schematic of atom arrangements in centrosymmetric and asymmetric FGT along c axis. **b** Observation of Fe_{II} atom deviation. The region is selected from a atomic-resolution HAADF-STEM image (c). **d** Line profile of the image intensity shown in (b). The blue and red lines represent the central position of two Te atoms and Fe_{II} atom, and their distance is estimated at 0.09

\AA , respectively.

Fig. R8 The results of first-principles calculations. **a** Schematic of DMI in FGT. The orange arrow D_1 shows the direction of DMI vector induced by Fe_{II} and top Te atoms, while the green arrow D_2 represents the opposite direction of DMI vector induced by Fe_{II} and bottom Te atoms. D_1 is not equal to D_2 , leading to a net DMI D_{eff} . **b** Spin configurations of counter-clockwise (CCW) (left column) and clockwise (CW) (right column). **c** Fe_{II} atom deviation-dependent D obtained by first-principles calculations.

4. In previous laws, the number of skyrmions generally increases, then decreases, and eventually disappears when a perpendicular magnetic field is applied (Appl. Phys. Lett. 2022, 121, 202402). However, it is intriguing to note that in Figure 3(a-f), the addition of a vertical magnetic field leads to a gradual decrease in the number of skyrmions. Furthermore, in Figure S6, there is an initial increase followed by a subsequent decrease. Therefore, what could be the inner reason for this discrepancy between these two samples?

Response: We sincerely thank the reviewer for careful reading of our manuscript. We agree with the reviewer that, in most cases, the number of skyrmions typically increases, subsequently decreases, and ultimately vanishes under the influence of the magnetic field. However, these observations are predicated on the premise that stripe domains constitute the magnetic ground state. This means that with an increase in the magnetic field, magnetic skyrmions gradually emerge from stripe domains and eventually annihilate, transitioning to the ferromagnetic state. In instances where the skyrmion domain constitutes the ground state, the number of skyrmions demonstrates a monotonically decreasing trend as a function of the magnetic field, as illustrated in Fig.

3 of this study and in Ref. 32-33.

Regarding the observation in Fig. S6, it is necessary to mention that the initial state at zero magnetic field (Fig. S6a) was in fact the remanent state, which had undergone both positive and negative applications of the magnetic field. This represents a metastable state, rather than a stable one. Fig. R9a displays the magnetization history of magnetic domain in Fig. S6 of the original manuscript. Therefore, it is logical to observe an initial increase followed by a subsequent decrease in the number of the skyrmions as the magnetic field intensifies. Consequently, we have modified the original Fig. S6 and its accompanying description in the revised version of Supplementary information (Page 5, Lines 8-10 and Lines 13-16), as also shown below:

“It should be noted that the FGT lamella underwent a field-swapping process, with its magnetization history depicted in Fig. S6a.”

“The reason can be attributed to the fact that the initial state at 0 mT represents a remanent state, indicative of a metastable state rather than a stable one. Consequently, observing a non-monotonic variation in skyrmion numbers is reasonable, which diverges from the trend elaborated upon in Fig. 3a.”

Fig. R9 The new version of Fig. S6 in the revised supplementary information.

5. The micromagnetic simulations are widely used to probe the topological magnetic texture of magnetic materials. In Fig.3(a-f), a phase diagram should be provided to explore the potential trend of the skyrmion number of the two samples and determine

an boundary among the different magnetic phases.

Response: We sincerely thank the reviewer for the valuable comments. Following the comments, the magnetic field-dependent skyrmion density and magnetic phases based on the LTEM observations in the FGT lamella are summarized, and the Fig. 3 in the original manuscript has been updated in revised manuscript, as shown in Fig. R10. Besides, we have added the following illustration of Fig. 3f,l to the revised version (page 23):

“The skyrmion density as a function of magnetic field at 0° (f) and 15° (l) tilt.”

Fig. R10 The new version of Fig. 3 in the revised manuscript.

6.The observation of both Bloch and hybrid skyrmions is interesting. However, what cause the mixing of a skyrmion with two compounds? Could it be attributed to crystal defects or impurity adsorption in the material?

Response: We appreciate the reviewer’s insightful question. In fact, the observation of mixed Bloch-Neel topological spin textures has also been reported in a previous study

about Co/Pd multilayers ([Phys. Rev. Lett. 122, 237201 (2019)] (ref. 36)). The reason that they declared was attributed to the energy balance between the exchange energy and DMI energy. Following the question, we have further carried out the micromagnetic simulations to better understand the physical mechanism behind it, as shown in Fig. R11.

The initial magnetic domain structures were configured as skyrmions domain, with DMI strengths varying from 0 to 0.8 mJ/m². Subsequently, the DMI strength for each skyrmion domain was gradually increased, and the corresponding total and exchange energies of the systems were obtained and summarized. The results indicate that for Bloch and hybrid skyrmions with a small *D* value, an appropriate increase in DMI strength results in only slight changes to the total and exchange energies, as depicted in the boxed region of Fig. R11. However, for skyrmions with a large *D* value near 0.8 mJ/m², these energies would undergo drastic changes as the DMI strength varied. Therefore, it is theoretically plausible to achieve mixed skyrmion phases, including Bloch and hybrid skyrmions, with *D* values near or even below 0.4 mJ/m², based on their minimal energy difference.

Drawing on both experimental and theoretical evidence, we posit that the emergence of mixed Bloch-Neel topological spin textures stems from the subtle competition between the magnetic dipolar interaction and DMI, and that a low DMI strength is essential to prevent the annihilation of Bloch skyrmions. Moreover, numerous recent studies have linked the DMI to the presence of Fe vacancies and defects in FGT crystals, as evidenced by findings in [Nat. Commun. 15, 1017, (2024)] and [Adv. Mater. 2108637, (2022)]. Indeed, as detailed in our response to comment #3, the presence of off-centered Fe_{II} atoms in FGT can induce an appropriate DMI. Consequently, it is reasonable to attribute the observation of mixed-type skyrmions in FGT, to some extent, to crystal defects. We anticipate that our findings will inspire further investigations into the distinct skyrmion phases in 2D ferromagnets.

Fig. R11 DMI-dependent magnetic total energy (a) and exchange energy (b) for skyrmions with various degree of hybridization (0 - 0.8 mJ/m²) in micromagnetic simulations.

7. The experimental results and micromagnetic simulation results exhibit excellent agreement, indicating a strong correlation between the data. How are the parameters of micromagnetic simulations derived from the experiments, and what is their underlying basis? If possible, the authors can make some relevant first-principle calculations.

Response: We thank the reviewer's comments on micromagnetic simulations. The Hamiltonian of the FGT in this work can be expressed as mainly includes exchange interaction, DMI, magnetic anisotropy, Zeeman, and stray field contributions, as follows:

$$H = E_{tot} = \int_v (E_{ex} + E_D + E_k + E_d + E_z) d\mathbf{r} = \int_v [A|\nabla\mathbf{m}|^2 + D\mathbf{m} \cdot (\nabla \times \mathbf{m}) + K_u(\mathbf{u} \cdot \mathbf{m})^2 - \frac{1}{2}M_s\mathbf{B}_d\mathbf{m} - B\mathbf{m} \cdot \hat{e}_z] d\mathbf{r} \quad (1)$$

Where E_{ex} , E_D , E_k , E_d , and E_z represent Heisenberg exchange energy, DMI energy, uniaxial anisotropy energy, demagnetization energy and Zeeman energy terms, respectively. Here, \mathbf{m} represents a normalized spin and \mathbf{u} is a unit vector of magnetic anisotropy. A , D , K_u and M_s are the exchange stiffness constant, DMI coefficient, uniaxial anisotropy constant and saturation magnetization, respectively. These magnetic parameters are set based on the bulk FGT crystal at room temperature. For

instance, the values of K_u and M_s were determined to be $0.8 \times 10^5 \text{ J.m}^{-3}$ and $2.4 \times 10^5 \text{ A.m}^{-1}$, respectively, by measured using a superconducting quantum interference device (MPMS (SQUID) VSM) magnetometer (Quantum Design Co.), as depicted in Fig. R12a-c. Additionally, estimating the exchange constant A requires the value of domain wall width, calculated using the formula: $A = k_u \left(\frac{w}{\pi}\right)^2$, where w and K_u are the domain wall width and uniaxial anisotropy constant, respectively. However, obtaining an accurate value for the domain wall width in the defocused mode of LTEM proved challenging, as the width varied with the defocusing amount. Therefore, we set A as an appropriate value of 5 pJ.m^{-1} based on the recent studies on FGT (1.5 pJ.m^{-1} for [Nat. Commun. 15, 1017, (2024)], 10 pJ.m^{-1} [Adv. Mater. 2311022, (2024)], 15 pJ.m^{-1} in [Adv. Mater. 2311022, (2024)] and 5 pJ.m^{-1} in [Adv. Sci. 10, 2303443 (2023)]). Furthermore, as detailed in our response to comment #3, the DMI strength has been determined to be below 0.94 mJ/m^2 through first-principles calculations, as illustrated in Fig. R12d. This value might be slightly larger than the actual one in our experiments, which is attributed to the omission of some realistic factors such as temperature and nonuniformity due to DFT's limitations. It should be highlighted that the estimated value of D ($< 0.6 \text{ mJ/m}^2$) obtained through micromagnetic simulations also aligns within the range of DFT-calculated values, underscoring the credibility of our findings.”

Fig. R12 The determination of magnetic parameters for FGT.

8. Can the authors provide magnetotransport measurements of 2D room-temperature ferromagnet Fe₃GaTe₂ to understand the transport characteristics of mixed-type skyrmions?

Response: We sincerely appreciate the insightful comments provided by the reviewer. Following the comments, we have further carried out the magnetotransport measurements of FGT, as shown in Fig. R13. A Hall-bar device with a FGT thickness of about 19.58 μm was fabricated (see Fig. R13a), and electric transport measurements over the temperature range of 300 - 100 K were implemented (see Fig. R13b-d). The results show that there is a discrepancy at the low-field region in magnetic hysteresis of Hall resistivity ρ_{xy} , demonstrating the presence of a pronounced THE component (see Fig. R13b). By linearly fitting the ρ_{xy} , we obtained the field dependent extracted topological Hall resistivity ρ_{xy}^T at various temperature (see Fig. R13c-d). It is evident that the THE signals are remarkable at 100 K and the maximum value of ρ_{xy}^T reach to 1.46 $\mu\Omega \text{ cm}$. With the increment of temperature, the ρ_{xy}^T reduces monotonically but

even apparent at 300 K with a maximum value of 0.73 $\mu\Omega$ cm. It is worth to mention that the topological Hall effect (THE) is indeed considered as a hallmark for the identification of nonlinear spin textures. Specifically, various spin structures with different topological charges will generate distinct signs of THE, e.g., the positive and negative THE induced by skyrmions and antiskyrmions, respectively. Nevertheless, to our knowledge, the hybrid skyrmions that observed by LTEM should have the same topological charges as the Bloch skyrmions. Therefore, it is reasonable that THE signals only show the integral topological property of the nonlinear spin textures. In other words, the individual identification of Bloch and hybrid skyrmions by THE signals is challenging. Even so, the existence of such broad-temperature THE signals show good agreement with the observations of high-thermostability topological skyrmions by LTEM.

In the revised version, we have added the associated discussions on transport properties on Page 12, Lines 9-12 in the main text and Page 9, Lines 9-20 in the Supplementary information, as also shown below:

“Importantly, such high-thermostability magnetic phases could also display robust topological Hall effect (THE) over a broad temperature, which is pivotal for their electric detection in topology-based memories, as illustrated in Fig. S12.”

“To investigate the potential THE in FGT, we fabricated a Hall-bar device with a sample thickness of about 19.58 μ m for the electric transport measurements over the temperature range of 300 -100 K, as shown in Fig. S12. Fig. S12a shows the schematic of the FGT Hall device with $I \parallel ab$ plane and $B \parallel c$ axis. The results show that there is a discrepancy at the low-field region in magnetic hysteresis of Hall resistivity ρ_{xy} , demonstrating the presence of a pronounced THE component (see Fig. S12b). By linearly fitting the ρ_{xy} , we obtained the field dependent extracted topological Hall resistivity ρ_{xy}^T at various temperature (see Fig. S12c-d). It is evident that the THE signals are remarkable at 100 K and the maximum value of ρ_{xy}^T reach to 1.46 $\mu\Omega$ cm. With the increment of temperature, the ρ_{xy}^T reduces monotonically but even apparent at 300 K with a maximum value of 0.73 $\mu\Omega$ cm. The existence of such broad-temperature

THE signals shows good agreement with the observations of high-thermostability topological skyrmions by LTEM.”

Fig. R13 Transport properties of FGT nanoflake. **a** Schematic of the FGT Hall device with $I \parallel ab$ plane and $B \parallel c$ axis. **b** Temperature-dependent Hall resistivity ρ_{xy} . **c** Representative extraction of topological Hall resistivity ρ_{xy}^T at 300 K. ρ_{xy}^N and ρ_{xy}^A shows the ordinary and anomalous Hall resistivity. **d** Temperature-dependent topological Hall resistivity ρ_{xy}^T . Blue and red curves were measured with decreasing and increasing magnetic field, respectively.

9. The scale bars of the graphs, such Fig.2(k,l), are not illustrated.

Response: We sincerely thank the reviewer for catching these mistakes. All the scale bars of the graphs have been checked and illustrated in revised manuscript now.

Once again, we would like to thank the reviewers for their time spent preparing their valuable feedback on our work. We hope that our responses to their comments and updates to the manuscript and well received.

REVIEWERS' COMMENTS

Reviewer #1 (Remarks to the Author):

The authors have addressed all the questions and the paper can be accepted now.

Reviewer #2 (Remarks to the Author):

The authors have addressed all my concerns, and the manuscript has been greatly improved. Therefore, I recommend the publication of this work with the current form.

Reviewer 1:

The authors have addressed all the questions and the paper can be accepted now.

Response: We sincerely thank the reviewer for recommending our manuscript to be published in Nature Communications.

Reviewer 2:

The authors have addressed all my concerns, and the manuscript has been greatly improved. Therefore, I recommend the publication of this work with the current form.

Response: We sincerely thank the reviewer for recommending our manuscript to be published in Nature Communications. The valuable suggestions and comments furnished by the referee have significantly contributed to enhancing the quality of our manuscript.